# Effects of rumen-protected methionine supplementation on the performance of high production dairy cows in the tropics

Valdir Chiogna Junior[1], Fernanda Lopes[2], Charles George Schwab[3], Mateus Zucato Toledo[4], Edgar Alain Collao-Saenz[1]*

1 Department of Animal Science, Federal University of Jataí, GO, Brazil, 2 Adisseo South America, São Paulo, SP, Brazil, 3 Schwab Consulting LLC, Boscobel, WI, United States of America, 4 Department of Dairy Science, University of Wisconsin, Madison, WI, United States of America

* edgar_collao_saenz@ufg.br

**Data Availability Statement:** All relevant data are within the manuscript and its Supporting Information files.

## Abstract

Increasing methionine availability in dairy cow diets during the first third of lactation may enhance their performance and health. The aim of this study was to determine the effect of supplementing rumen-protected methionine (Smartamine® M, SM) in a lactation diet with protein and energy levels calculated according to the literature. Seventy-six multiparous Holstein cows (39.1 ± 6.8 kg of milk/d and 65 ± 28 DIM) were assigned to 1 of 2 dietary treatments (38/treatment) according to a randomized complete block design with a 2-wk (covariate) and 10-wk experimental period. Treatments were a basal diet (CON; 3.77 Lys:1Met); and CON + 23 g SM (2.97 Lys:1 Met). Individual milk samples were taken every 2 weeks to determine milk composition. Blood was collected from 24 cows on d+30 d to measure plasma AA levels. Body weight and body condition score (BCS) were measured at the beginning and the end of the experiment. The SM diet promoted higher milk yield (41.7 vs. 40.1 kg/d; $P = 0.03$). Energy-corrected milk yield (41.0 vs. 38.0 kg/d), milk protein yield (1.30 vs. 1.18 kg/d), milk protein (3.14% vs. 2.97%) and casein (2.39% vs. 2.28%) were also different ($P < 0.01$) as well as milk fat yield (1.42 vs. 1.29 kg/d; $P = 0.02$). A trend ($P = 0.06$) for higher milk fat % (3.41% vs. 3.21%) was observed. Both diets resulted in similar body weight, but CON-fed cows tended ($P = 0.08$) to have higher BCS. Higher plasma methionine levels were determined with SM compared with CON (29.6 vs. 18.4 µM; $P < 0.01$), but lysine and histidine were not different. Dietary supplementation of RPM improved productive performance by increasing milk yield and milk components yields, suggesting better dietary AA utilization when Met levels are adjusted in Lys-adequate lactation diets.

## Introduction

Precision feeding is a valuable strategy to improve income-over-feed-cost in dairy production and feeding AA-balanced diets may be an effective strategy to increase milk yield and milk components [1]. Dairy nutritionists are changing their focus from balancing rations for crude

**Funding:** This research was supported by Adisseo SA Inc. (São Paulo, SP; adisseo.com), and Schwab Consulting LLC. Boscobel, WI, USA. The funder provided support in the form of salaries for authors Lopes, F Schwab CG, but did not have any additional role in the study design, data collection and analysis, decision to publish, or preparation of the manuscript. Dr. Fernanda Lopes and Dr. Charles G. Schwab helped with designing the diets and were involved in discussion of results.

**Competing interests:** Dr. Fernanda Lopes is an employee of Adisseo SA Inc. and Dr. Charles G. Schwab works for Schwab Consulting LLC. Other authors have declared that no competing interests exist. This does not alter our adherence to PLOS ONE policies on sharing data and materials.

protein (CP) to rumen degradable protein (RDP) and more optimum concentrations of AA in rumen undegradable protein (RUP); the latter of which provides for more optimum concentrations of AA in metabolizable protein. These changes are being implemented to better meet but not exceed the protein requirements of ruminal microorganisms and the metabolizable AA requirements of the animals. The goal is to reduce the need for supplemental protein and reduce the risk of the most limiting AA limiting animal performance. The desired outcome is reduced feed costs, increased milk, and milk component yields, improved dietary protein efficiency, and reduced environmental pollution [2].

Lysine (Lys) and methionine (Met) have been identified most frequently as the two most limiting AA for milk protein synthesis. This is not surprising given their low content in most feed proteins relative to their content in mixed rumen microorganisms and milk protein. In vitro studies using mammary epithelial cells have also shown the importance of Lys and Met concentrations in the medium, not only on maximal milk protein synthesis but also on AA transport and signal transduction pathways affecting expression of genes related to milk protein synthesis [3]. While the data indicated that both AA can activate the expression of genes involved in milk protein transcription and translation, peak concentrations of casein and cell proliferation rates were observed when the ratio of supplemental Lys and Met was ~3:1. The optimum concentrations for Lys and Met in metabolizable protein have been defined for different nutritional models [4, 5] and have proven useful for balancing diets for these two AA.

Methionine has long been recognized as being the first limiting AA for lactating dairy cows, particularly when fed high Lys, low Met-containing protein supplements such as blood meal and soybean meal. An effective approach to increase post-ruminal Met supply is feeding rumen-protected Met (RPMet). Several studies have shown the beneficial effects of RPMet supplementation on sparing of dietary protein [6] and overall lactation performance of dairy cows when diets were based on typical North American ingredients dairy cow performance [6, 7]. A recent focus has been on RPMet supplementation of transition cow diets. Met supplementation to achieve a Lys:Met ratio of approximately 2.9:1 as % of MP improved milk yield, which was partially attributed to an increase in dry matter intake (DMI) and possible better utilization of body lipid reserves [7]. Those authors also reported enhanced immune function and suggest that these results support the fine-tuning of EAA content as a percentage of MP requirement in diet formulation.

Methionine is the only AA which contains sulfur acting as a precursor for other sulfur-containing AA (SAA), such as cysteine, homocysteine, and taurine, which are important in methylation reactions. Methionine metabolism starts with its conversion into S-adenosylmethionine (SAM), which is a key cofactor of the Met intermediate metabolism used for methylation reactions. SAM is a methyl donor to a wide variety of acceptors, such as amino acid residues in proteins, DNA, RNA, and small molecules [8]. In addition to SAM, methionine acts as precursor of hydrogen sulfide, taurine, and glutathione. These products are reported to alleviate oxidant stress induced by various oxidants and protect the tissue from the damage [9]. Therefore, dietary supplementation of rumen-protected methyl donors, such as RPMet, may allow to meet the requirements of cows at the peak of lactation, when the output of methylated compounds in the milk is high.

Most continuous feeding studies investigating the impacts of Met-supplementation of otherwise nutrient balanced diets have been carried out under research-controlled conditions during the transition period or beginning at the onset of lactation. There are limited data on dietary Met supplementation around the peak of lactation under field conditions and its effects during the remaining of the lactation, and in particular when cows are maintained under tropical conditions. The hypothesis of this experiment was that supplementing a corn-based, soybean meal lactation diet with RPM to achieve the desired 3.0:1 Lys:Met ratio in MP [4] would

improve the performance of lactating cows during the first half of lactation and can affect the plasma AA profile. The objectives were to measure milk yield and milk component responses, including peak milk yields, of mid-lactation dairy cows, by supplying RPMet to a basal diet formulated for high milk production. Plasma free AA were also measured to confirm that the RPMet supplement provided the intended supply of metabolizable Met.

## Materials and methods

### Animal housing and care

The experiment was conducted on a commercial dairy farm in the state of Goias, in the central-western region of Brazil. All experimental procedures were approved by the Committee of Ethics on Animal Use (CEUA) of the Federal University of Goias–Campus Jataí, under protocol number 024/17.

Although the weather conditions in southwest of Goias State are not extremely hot and humid during the time when this work was conducted. In 40% of the experiment days, the average local temperature-humidity index exceeded 72, considered the critical point at which milk yield is reduced, and the minimum and maximum THI averaged 61.0 and 74.2. Maximum THI exceeded 72 in more than 80% of days of the trial period. Cows were housed in two separate single pens, one for each treatment, in a free-stall barn with sand bedding and a cooling system (sprinklers and fans) to provide good animal welfare conditions. Cows had free access to drinking water. A total mixed ration (TMR) was offered twice daily (at 06:00 and 15:00 h). Feed allowance was adjusted daily to ensure 5% feed refusals. Cows were milked three times daily at 05:30, 12:00, and 20:00 h.

### Experimental design and dietary treatments

Seventy-six lactating multiparous Holstein cows with a previous lactation average milk yield of 10600 kg were distributed evenly to one of two experimental diets (Table 1) according to a randomized complete block experimental design (38 blocks). Blocking criteria were milk yield and days-in-milk (DIM). At the beginning of the study, cows had an average of 2.75 ± 1 lactations, 645.4 ± 45.6 kg BW, and 3.3 ± 0.3 body condition score (BCS). The duration of the experiment was 12 weeks: the first two weeks were used for diet adaptation (covariate), and the following 10 weeks were considered the experimental period.

**Table 1.  Ingredient and nutritional composition of treatments diets.**

|  | Diet[1] | |
|---|---|---|
|  | **CON** | **SM** |
| *Average at the beginning of the experiment* |  |  |
| Milk Yield (kg) | 38.9 ± 7.1 | 39.2 ± 6.7 |
| BW (kg) | 650 ± 46 | 640 ± 45 |
| BCS | 3.32 ± 0.3 | 3.26 ± 0.3 |
| DIM | 64 ± 27 | 66 ± 30 |
| *Ingredient (% of DM)* |  |  |
| Corn silage[2] | 45.81 | 45.81 |
| Fresh grass (Tifton)[3] | 4.41 | 4.41 |
| Soybean meal, solv.[4], 46% CP | 15.57 | 15.57 |
| Soybean meal treated[5] | 1.25 | 1.25 |
| Smartamine® M[6] | - | 0.09 |
| High-moisture corn[7] | 17.28 | 17.28 |

(*Continued*)

**Table 1.** (Continued)

| | Diet[1] | |
| --- | --- | --- |
| | **CON** | **SM** |
| Soybean hulls | 10.36 | 10.36 |
| Saturated fat supplemented[8] | 1.81 | 1.81 |
| Urea | 0.23 | 0.23 |
| Mineral-vitamin premix[9] | 3.20 | 3.11 |
| *Nutritional composition[10], %* | | |
| Crude protein | 16.7 | 16.7 |
| RDP | 10.3 | 10.3 |
| RUP | 6.4 | 6.4 |
| NDF | 30.8 | 30.8 |
| Forage NDF | 20.8 | 20.8 |
| ADF | 19.0 | 19.0 |
| NFC | 42.3 | 42.3 |
| Starch | 27.7 | 27.7 |
| Ether extract | 4.6 | 4.6 |
| Net energy for lactation (NEl, Mcal/kg) | 1.6 | 1.6 |
| Ca | 1.0 | 1.0 |
| P | 0.4 | 0.4 |
| *Dietary Protein Balance[10]* | | |
| RDP required, g/d | 2595 | 2597 |
| RDP supplied, g/d | 2711 | 2715 |
| RDP balance, g/d | 116 | 118 |
| RUP required, g/d | 1539 | 1537 |
| RUP supplied, g/d | 1672 | 1688 |
| MP balance, g/d | 113 | 129 |

[1]CON = Control diet; SM = CON + SM (0.09% of DM)

[2] Corn silage = 33.5% DM, 7.98% CP, 33.3% starch, 39.5% NDF

[3]Fresh grass = 28.8% DM, 14.9% CP, 60.7% NDF

[4]Soybean meal = Soybean meal 46% CP Cargill® Inc.

[5]Soybean meal, treated = Soy Pass® BR Cargill® Inc.

[6]Smartamine® M = rumen-protected methionine, Adisseo Inc., Antony, France

[7]High moisture corn = 64.8% DM, 8.8% CP, 69% starch

[8]Saturated fat supplemented = Enerfat®, Kemin® Industries Inc., USA

[9]Mineral-vitamin premix = Precisão Nata® (Campo® Rações e Minerais); Mg 35 g/kg, Na 85 g/kg, Cl 80 g/kg, Ca 190 g/kg, P 24 g/kg, S 4,000 mg/kg, K 9.900 mg/kg, Cu 450 mg/kg, Zn 2,300 mg/kg, Fe 2,000 mg/kg, Mn 1,800 mg/kg, I 25 mg/kg, Co 30 mg/kg, Se 15 mg/kg, F 300 mg/kg, Vit. A 156,000 IU/kg, Vit. D 51,500 IU/kg, Vit. E 1,000 IU/kg, Biotin 35 mg/kg, Cr 20 mg/kg, Monensin 510 mg/kg, *Saccharomyces cerevisae* 1.40x10⁵ CFU/kg.

[10]Predicted using NRC (2001) and laboratory analysis of the feeds.

The two experimental diets contained equal nutrient levels, except for Met. The basal diet was formulated using the NRC (2001) model [4] to meet the cows' nutritional requirements for an average milk yield of 43 kg/d, an average BW of 645 kg, and an average milk composition of 3.6% fat and 3.07% true protein. Model-predicted dry matter intake was 26.3 kg/d. SMARTAMINE® M (Adisseo Inc., Antony, France) is a lipid/pH-sensitive polymer-protected product that contains 75% DL-Met with an assumed Met bioavailability value of 80% [10]. Smartamine M was pre-mixed with the mineral/vitamin premix, urea, treated soybean meal and saturated fat and then mixed in the TMR.

**Table 2. Predicted flows of EAA to the small intestine (NRC, 2001)[1].**

| | Diet[2] | | | | | |
| | CON | | | SM | | |
| Amino acid | Flow (g/d) | DigAA flow (g/d) | % of MP | Flow (g/d) | DigAA flow (g/d) | % of MP |
|---|---|---|---|---|---|---|
| Arginine | 169 | 141 | 4.79% | 169 | 142 | 4.77% |
| Histidine | 75 | 62 | 2.12% | 75 | 62 | 2.10% |
| Isoleucine | 173 | 142 | 4.81% | 173 | 142 | 4.79% |
| Leucine | 312 | 256 | 8.67% | 312 | 256 | 8.62% |
| Lysine | 228 | 188 | 6.37% | 229 | 189 | 6.35% |
| Methionine | 61 | 50 | 1.69% | 75 | 64 | 2.14% |
| Phenylalanine | 179 | 147 | 4.98% | 179 | 148 | 4.97% |
| Threonine | 171 | 140 | 4.73% | 171 | 140 | 4.71% |
| Valine | 194 | 158 | 5.37% | 194 | 159 | 5.35% |
| Total | 1562 | 1286 | 43.53% | 1578 | 1301 | 43.8% |

[1]The NRC (2001) evaluation of diets was based on an expected average milk yield of 43 kg/d, an average BW of 645 kg, an average milk composition of 3.6% fat and 3.07% true protein and a DMI of 26.3 kg/d

[2]CON = Control diet,Lys:Met ratio 3.77:1; SM = CON + SM (0.09% of DMI), Lys:Met ratio 2.97:1

The nutritional composition of the experimental diets and predicted protein balances are shown in Table 1, calculated according to the NRC (2001). Predicted flows of EAA to the small intestine (NRC, 2001) are shown in Table 2. Model-predicted flows of MP and EAA indicated that 14 g/d of additional metabolizable Met was needed to achieve an approximate predicted Lys/Met ratio of 3.00/1 in metabolizable protein, With a minimum content of 75% DL-Met and an assumed Met bioavailability value of 80%, this required 23/d of SMARTAMINE® M.

## Sampling and measurements

Feed allowance and refusal amounts were recorded daily to calculate the group feed intakes. Samples of the TMRs were collected twice weekly to determine DM content (%). Forages were analyzed for DM content every 15 days to adjust diet allowance. Composite feedstuff samples (corn silage, fresh grass Tifton and high moisture corn) were analyzed by wet chemistry in the 3rLab (Lavras, MG, Brazil) for CP, neutral detergent fiber (NDF), acid detergent fiber (ADF), starch, ether extract (EE), ash, calcium (Ca), phosphorus (P) contents according to the methods of the AOAC International.

Composite milk samples were made for each cow from the 3 milkings of the same day once during the adaptation period and every two weeks until the end of the study. Samples were preserved in 2-bromo-2nitropropane-1,3-diol, and submitted to the milk lab of the University of São Paulo (Clínica do Leite, USP, Piracicaba, SP, Brazil), where infrared analyses of total solids content, milk urea nitrogen (MUN), and somatic cell count (SCC) were performed (ISO 9622:2013/IDF 141:2013).

Cows were milked three times daily at 05:30, 12:00, and 20:00 h. Milk yield was individually recorded using a dairy management software (DairyPlan®). Energy-corrected milk yield (ECM) was calculated as ECM = [(0.323 x kg milk) + (12.82 x kg fat) + (7.13 x kg true protein)]. During the adaptation period and at the end of the experimental period, cows were individually weighed and their body condition scores determined after the second milking of the day.

Blood samples were collected from 12 cows on each treatment on day 30 of the experimental period (DIM = 89 ± 27 SD) to measure plasma AA levels. Blood was collected from the tail

vein after the second milking, about 6 hours after the 1[st] feeding. Blood plasma was immediately separated by centrifugation at 1.500 x g for 15 min, aliquoted into 2-mL tubes, and frozen at -20˚C until analyses at the Dairy Science Department, University of Wisconsin-Madison, USA. Briefly, plasma samples were combined with an internal standard (homoarginine, methionine-d$_3$, and homophenylanine) before deproteinization with 1 N perchloric acid (final concentration 0.5 N) and the supernatant collected and filtered Rhrough a 0.22 μm filter (Millex-GV filter; Millipore Sigma, Burlington, MA). Filtered samples were separated using a 50 x 3 mm column (length x i.d; mixed mode with normal phase plus ion exchange; Intrada Amino Acid, Imtakt Inc., OT, USA) and the Nexera-i LC-2040C (Shimadzu, Kyoto, Japan). Mobile phases were acetronile containing 0.1% of formic acid (A) and 100 m$M$ ammonium formate in water (B), with 14% B gradient for 0–3 min, increasing to 100% B gradient for 3–10 min, followed by a decreased to 14% for 10–12 min and re-equilibrated with 14% B gradient for 12–16 min. The eluent was ionized using electrospray ionization and analyzed in positive selected ion-monitoring mode using a Single quadrupole mass spectrometry (LCMS-2020; Shimadzu, Kyoto, Japan. Samples were analyzed in duplicates

## Statistical analysis

Data were analyzed according to a randomized complete block design using the MIXED procedure of SAS statistical package (version 9.4, SAS Institute Inc., Cary, NC, USA). The model included a covariate (data collected during a 2-wk adaptation period), block, treatment, week, and the relevant interactions as fixed effects, and cow within block as random effects. Week of treatment was included as a repeated measure using the autoregressive of order 1 covariance structure to account for autocorrelated errors. Means were calculated using the least squares means statement, and treatment means were compared using Bonferroni t-test when overall treatment F-test was significant ($P<0.05$). Interaction effects were partitioned using the SLICE options of SAS (version 9.4, SAS Institute Inc.). Statistical significance and trends were considered at $P \leq 0.05$ and $P > 0.05$ to $P \leq 0.10$, respectively. Plasma AA levels (μM) were analyzed and compared between treatments as described above but excluding week and treatment x week interaction.

$$Y_{ij} = \mu + c_i + T_j + e_{ij},$$

where $Y_{ij}$ = dependent variable, $\mu$ = overall mean, $c_i$ = random effect of cow ($i$ = 1 to 24), $T_j$ = fixed effect of treatment ($j$ = 1 or 2), and $e_{ij}$ = residual error.

## Results and discussion

The typical recommendations for balancing dietary AA are made relative to total MP supply. The SM diet was formulated to ensure an adequate amount of Met in MP, relative to the predicted concentration of Lys in MP. The NRC [4] suggests the required predicted concentrations of Lys and Met in MP for maximal milk protein yield are 7.2% and 2.4%, respectively. As these amounts are difficult to achieve without both RPLys and RPMet supplementation, it is recommended that the first step to balancing diets for Lys and Met is to feed high-Lys protein supplements like blood and soybean meal and then supplement with RPMet to maintain their ratio in MP at 3:1 while trying to achieve practical levels in MP as close as possible to 6.6% and 2.2%, respectively [11]. In this experiment, the predicted concentrations of Lys and Met in MP in the CON diet were 6.37 and 1.69%, respectively. Prior evaluation of the CON diet with NRC (2001) indicated that 23 g/d of Smartamine M was needed to achieve a predicted Lys:Met ratio in MP of 3:1. The predicted concentrations of Lys and Met in MP in the SM diet were 6.35 and 2.14%, respectively.

The predicted ratios of metabolizable Met and Lys relative to predicted supplies of metabolizable energy (ME) were 1.17 g Met/Mcal ME and 3.05 g Lys/Mcal ME in the SM diet. These values are close to the suggested optimum values of 1.14 g Met/Mcal and 3.03 g Lys/Mcal [12]. Using the latest version of the CNCPS software [5], predicted concentrations of Lys and Met in MP were 6.57 and 2.45, respectively, yielding a Lys:Met ratio in MP of 2.7:1. The required predicted concentrations for Lys and Met in MP for maximal milk protein yields are 7.00 and 2.60%, respectively [5]. This provides for an optimal Lys/Met ratio in MP of 2.69/1.

In the present experiment, the DMI average of the group-fed cows in each treatment was not affected ($P>0.05$) by treatment (26.2 kg/d for CON vs. 26.1 kg/d for SM). Previous studies on RPMet supplementation reported inconsistent DMI responses. While some studies did not find any effect on DMI [6, 13–16], others observed an increase in DMI [7, 17–19]. Most of the studies reporting an increase in DMI were initiated before or at the time of calving [7, 17–19]. In these studies, the increases in DMI were partially attributed to reduced inflammation and oxidative stress, resulting in improvements in immunometabolic status and liver function [20, 21]. According to a meta-analysis of 35 studies [22], differences in the level of Met supplementation, presence of potentially co-limiting AA, length of feeding period, and stage of lactation may all affect feed intake and, therefore, confound DMI results.

Effects of dietary treatments on milk and milk component yields and milk composition are presented in Table 3. The SM diet promoted higher milk yield, ECM, FCM, milk protein percentage, milk protein yield, milk casein percentage, milk fat yield, and total solids yield, and tended ($P = 0.06$) to increase milk fat percentage (Table 3) than the CON diet. Milk lactose content was lower (4.54% vs. 4.61, $P = 0.03$) for cows on SM versus CON; the same tendency for lower lactose percentage was observed in other experiment [23]. Cows on CON tended to present higher BCS (3.4 vs. 3.3; $P = 0.08$) compared with the SM cows although no BW differences were detected (656.5 vs. 648.5 kg; $P = 0.12$). There was no influence of dietary treatment on milk lactose yield, total solids percentage, MUN, or SCC.

**Table 3. Effects of the dietary treatments on milk yield and composition in dairy cows.**

| | Diet[1] | | | P-value | | |
|---|---|---|---|---|---|---|
| Item | CON | SM | SEM | Diet | week | Diet x week |
| Milk yield, kg/d | 40.0 | 41.7 | 0.50 | 0.03 | 0.19 | 0.87 |
| ECM[2], kg/d | 38.0 | 41.0 | 0.67 | <0.01 | 0.07 | 0.99 |
| FCM[3], 3.5% | 38.3 | 41.1 | 0.77 | 0.01 | 0.14 | 0.97 |
| Milk fat, % | 3.21 | 3.41 | 0.07 | 0.06 | 0.85 | 0.99 |
| Milk fat, kg/d | 1.29 | 1.42 | 0.04 | 0.02 | 0.64 | 0.99 |
| Milk protein, % | 2.97 | 3.14 | 0.03 | <0.01 | <0.01 | 0.95 |
| Milk protein, kg/d | 1.19 | 1.30 | 0.01 | <0.01 | <0.01 | 0.95 |
| Milk casein, % | 2.28 | 2.39 | 0.03 | <0.01 | <0.01 | 0.96 |
| Milk lactose, % | 4.61 | 4.54 | 0.02 | 0.03 | <0.01 | 0.83 |
| Milk lactose, kg/d | 1.85 | 1.89 | 0.03 | 0.26 | 0.02 | 0.80 |
| Total solids, % | 11.78 | 12.10 | 0.13 | 0.10 | 0.42 | 0.96 |
| Total solids, kg/d | 4.75 | 5.01 | 0.09 | 0.05 | 0.20 | 0.96 |
| MUN, mg/dL | 10.43 | 10.74 | 0.24 | 0.37 | <0.01 | <0.01 |
| SCC, X10[3] | 135.86 | 180.50 | 44.70 | 0.48 | 0.13 | 0.37 |

[1]CON = Control diet; SM = CON + SM (0.09% of DMI)

[2]Energy-corrected milk (kg/d) = [(0.323 x kg milk) + (12.82 x kg fat) + (7.13 x kg true protein)] (Hutjens, 2010).

[3]Fat-corrected milk (3.5%) = 0.4318 kg milk + 16.23 kg milk fat

## Milk yield

Higher milk yield ($P = 0.03$), FCM ($P = 0.01$) and ECM ($P<0.01$) were observed in the SM treatment. The mechanisms by which RPMet supplementation increases milk yield are not fully understood. However, considering mechanisms proposed in other experiments [24, 25] who also observed improved lactation performance with metabolizable methionine supplementation, we suggest that the milk yield response in this experiment was due to increased efficiency of use of the other absorbed proteinogenic AA by the mammary gland for milk component synthesis. There are three reasons for this suggestion. The first is that the predicted supply of ME for the achieved milk yield was not limiting, hence there was an opportunity for increased milk yield without increased feed intake. The second factor is that milk yield is a function of milk component synthesis, particularly lactose and protein. It is well established that yields of lactose and protein are highly and positively correlated with milk yield. And third, in addition to being a substrate limiting AA for protein synthesis, it also possesses signaling properties, along with leucine, isoleucine and histidine, for promoting cellular anabolic metabolism. It has been shown to activate protein translation by being mechanistically linked to mTOR regulation [26]. The result, if this occurred in our study, would be increased efficiency of use of all AA for protein synthesis.

Fig 1 shows the lactation curves (average weekly milk yields), according to treatment during the 10-week experimental period. Most striking is how quickly the cows responded to Met supplementation. This observation indicates that Met was not only the first limiting AA, but also the most limiting nutrient, and that other aspects of dairy farm management were not limiting animal performance.

Several studies reported increases in milk yield when RPMet was supplemented to periparturient cows [6, 7, 17–19], but only a few started supplementation at or shortly after peak lactation. Such studies indicate that milk yield responses to Met supplementation during that period is influenced by several factors, including Met supply and source [6], adequacy of supplies of metabolizable Lys [24], MP and RDP in the diet and stage of lactation [27].

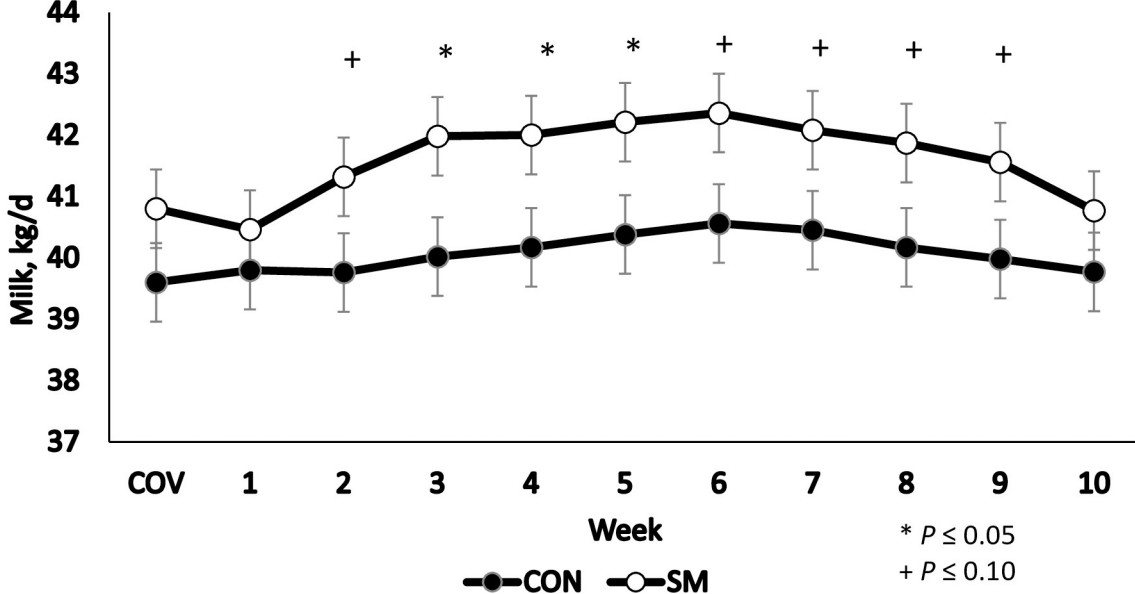

**Fig 1. Effect of dietary RPMet supplementation on milk yield compared with the control diet.** Values are expressed in average weekly milk yield (kg/d).

Although Met is a glycogenic AA, and most of the glucose required for lactose synthesis in the mammary gland is derived from gluconeogenesis [28], AA are not prioritized for glucose production, and the quantitative contribution of EAA to glycogenic carbon is very small [29]. Indeed, in cattle as well as other animals, Met has other metabolic priorities [30] for essential biological processes such as protein synthesis, DNA methylation, SAM-dependent trans-methylation reactions, polyamine formation, and synthesis of glutathione, phosphatidylcholine, and the non-essential AA cysteine [31]. It is assumed that the biological priority for these purposes exceeds Met utilization for hepatic gluconeogenesis [32].

Doepel and Lapierre [33] observed a positive response of the mammary gland of dairy cows to EAA infusion, increasing milk and milk protein yields, without a concurrent increase in glucose uptake but with an increase of the uptake of BHBA plus lactate, and concluded that this intensification of use of other energy-yielding substrates, demonstrates the metabolic priority of milk production and the flexibility of the mammary gland to use various substrates at its disposal to support higher milk and protein yields.

## Milk components

Independently of week, RPMet supplementation increased milk protein percentage ($P<0.01$), milk protein yield ($P<0.01$), and milk casein percentage ($P<0.01$). Amino acids not only serve as direct precursors for protein synthesis, but also act as regulators of protein synthesis rate. When activated by Met, mTOR may increase the initiation rates of protein synthesis, which may explain why lactating dairy cows fed corn-based diets typically increase milk protein production when supplemented with RPMet [34].

Although the amounts of absorbed EAA are important for milk protein synthesis, being absorbed in their correct balance is also essential for maximum protein yield [35, 36]. An in vitro experiment using bovine mammary cells indicated that the ratio of Lys to Met influences the expression of casein by bovine mammary epithelial cells [3] and, may explain the higher milk casein percentage observed in the present study. The peak of casein concentration in was observed at 3:1Met: Lys ratio, and appeared to be at least partially driven by the upregulation of mRNA expression and mTOR pathways [3]. Several studies evaluating RPMet supplementation to lactating cows reported increases in milk protein percentage [7, 13, 17, 18], and, according to Schwab et al. [11], the magnitude of the increase is directly proportional to the adequacy of Met in metabolizable protein.

Cows fed the SM diet had a higher yield of milk fat ($P = 0.02$), a result of higher milk yields and a trend of higher milk fat percentage ($P = 0.06$) compared with cows fed the CON diet (Table 3). The effect of Met supplementation on milk fat synthesis, with resulting increases in milk fat percentage, warrants further investigation; however, some mechanisms have been proposed. Methionine is considered a lipotropic agent. Possibly, the most important roles of Met in the liver is to stimulate synthesis of very low-density lipoprotein (VLDL) and as a methyldonor to enhance the synthesis of the methylated compound, S-adenosylmethionine (SAM) [37]. In turn, SAM can be used to produce phosphatidylcholine, which is the main component of the membrane that encloses VLDL, allowing their assembly and secretion from the liver [38]. The result is reduced accumulation of triacylglycerols (TAG) in the liver. About 40 to 60% of milk fat consists 18 and 16-carbon fatty acids, which are almost entirely derived from VLDL. In the mammary gland, VLDL is anchored to the endothelium by the enzyme lipoprotein lipase (LPL), which then hydrolyzes TAG present in the lipoprotein nucleus to release fatty acids [39]. The same authors concluded that Met may also influence milk fat synthesis by upregulating lipogenic gene networks and changing the expression of key miRNA involved in the control of lipogenic balance. This indicates a potentially important role of EAA ratios and mTOR signaling pathways in the regulation of milk fat synthesis [40].

Cows fed the CON diet showed higher milk lactose percentage ($P$ = 0.03) than the SM-fed group, but differences in milk lactose yield were not detected (Table 3), suggesting the dilution of this component in the higher milk yield obtained with the SM diet. The higher total solids yield ($P$ = 0.05) of the SM-fed cows, but the lack of significant difference of total solids percentage relative to those fed the CON diet may be because of the observed lower lactose percentage obtained with the SM diet.

## Plasma AA concentrations

Plasma free AA concentrations are reported in Table 4. There were significant effects of RPMet supplementation on plasma Met and alanine (Ala) concentrations ($P$<0.05).

Dietary RPMet supplementation increased plasma Met concentrations by 61% (29.6 vs. 18.4 μM; $P$ < 0.001). The increase in plasma Met concentrations was significantly higher than reported by others feeding similar amounts of supplemental Met from other RPMet supplements [23, 41–43]. Increases in plasma free Met in "Met-deficient" Holstein cows duodenally infused with incremental amounts of DL- Met (g/d) were reported as 1.25 μM of plasma Met/g of Met infused [44, 45]. Using the same approach, we found 11.2 μM of plasma Met increase for an intake of 17.25 g of fed DL-Met (0.65 μM of plasma Met/g of Met) and, a calculated estimate of Met bioavailability of 52% (0.65/1.25). This attests to the high bioavailability of Met in Smartamine M. The plasma Met value obtained with the CON diet is very similar to concentrations reported by others when cows are fed corn-based diets supplemented with high Lys protein supplements with RPMet supplementation [11, 46, 47]. Milk protein concentration increased in 0,17% in response to a 23 g supplementation of a product with 75% of MET content (17.25 g MET), the change was 0.01% / g of consumed Met. This value was almost the double of the 0,0047% /g reported in other experiment with the same product [41] suggesting that other variables in the diet, environment or cows can interact with this response.

**Table 4. Effect of dietary RPMet supplementation on plasma AA concentrations (μM) of dairy cows[1].**

|  | Diet[2] | | |
|---|---|---|---|
|  | **CON** | **SM** | **$P$-value** |
| Arg | 77.1 ± 4.8 | 86.0 ± 6.8 | 0.29 |
| His | 33.5 ± 3.8 | 31.7 ± 5.3 | 0.79 |
| Ile | 78.7 ± 5.2 | 70.0 ± 7.3 | 0.34 |
| Leu | 73.7 ± 7.5 | 70.0 ± 10.5 | 0.78 |
| Lys | 57.4 ± 4.5 | 64.7 ± 6.3 | 0.36 |
| Met | 18.4 ± 1.4 | 29.6 ± 1.9 | **<0.001** |
| Phe | 38.7 ± 1.6 | 39.0 ± 2.3 | 0.91 |
| Thr | 54.0 ± 3.6 | 62.1 ± 5.0 | 0.21 |
| Trp | 31.2 ± 1.7 | 32.3 ± 2.4 | 0.73 |
| Val | 220 ± 16.4 | 212 ± 22.9 | 0.76 |
| Ala | 206 ± 15.9 | 272 ± 22.3 | **0.03** |
| Asn | 24.0 ± 2.2 | 30.5 ± 3.1 | 0.10 |
| Gln | 95.4 ± 8.3 | 90.6 ± 11.6 | 0.74 |
| Glu | 63.4 ± 4.3 | 59.6 ± 6.0 | 0.61 |
| Pro | 72.9 ± 5.4 | 84.4 ± 7.5 | 0.23 |
| Ser | 104 ± 15.8 | 97.7 ± 22.2 | 0.80 |
| Tyr | 38.7 ± 4.4 | 39.1 ± 6.1 | 0.95 |

[1]Data are presented as mean ± SEM

[2] CON = Control diet; SM = CON + SM (0.09% of DMI)

Similarly, RPMet supplementation increased ($P$ = 0.03) plasma Ala level in 32% (206.3 μM vs 271.6; $P$ = 0.03). There are evidences of increased contribution of Ala to liver glucose synthesis at beginning of lactation. Since Ala is an essential glucogenic AA, and hepatic gluconeogenesis from Ala is greater than from other AA, the higher Met levels may have improved hepatic metabolism sufficiently to increase Ala release from the muscle into the bloodstream [48]. There was no influence of dietary treatment on plasma levels of the other AA.

## Conclusions

A positive response in milk yield and milk component was observed by supplying RPMet to a basal diet for mid-lactation dairy cows. We conclude that Met was limiting in this experiment and formulating rations for a more ideal Lys/Met ratio of absorbed AA was of significant benefit to the cows. These findings suggest that, when the diet contains more optimal adequate Lys levels, Met supplementation to achieve the correct Met to Lys ratio improves AA utilization. The best performance appeared to be associated with plasma methionine concentration close to 30μM.

Overall, the results of this experiment underscore the benefits of balancing for a more ideal profile of absorbed AA in lactating dairy cows on milk and milk component yields and overall production efficiency.

The increase in plasma Met concentrations indicate high bioavailability and stability of the rumen protected MET product used in this experiment even when mixed with other feeds or wet TMR.

## Supporting information

**S1 Table. Effects of the dietary treatments on milk yield and composition in dairy cows.**
(PDF)

**S2 Table. Effect of dietary RPMet supplementation on plasma AA concentrations (μM) of dairy cows1.**
(PDF)

**S1 Fig.**
(PDF)

## Author Contributions

**Conceptualization:** Valdir Chiogna Junior, Fernanda Lopes, Edgar Alain Collao-Saenz.

**Data curation:** Valdir Chiogna Junior.

**Formal analysis:** Charles George Schwab, Mateus Zucato Toledo, Edgar Alain Collao-Saenz.

**Funding acquisition:** Fernanda Lopes.

**Investigation:** Valdir Chiogna Junior, Edgar Alain Collao-Saenz.

**Methodology:** Valdir Chiogna Junior, Mateus Zucato Toledo, Edgar Alain Collao-Saenz.

**Project administration:** Fernanda Lopes, Edgar Alain Collao-Saenz.

**Supervision:** Fernanda Lopes.

**Visualization:** Valdir Chiogna Junior, Fernanda Lopes, Charles George Schwab.

**Writing – original draft:** Valdir Chiogna Junior.

**Writing – review & editing:** Charles George Schwab, Edgar Alain Collao-Saenz.

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
