## [Decision Letter · Decision Letter 0]

14 Jan 2021

PONE-D-20-37159

Effects of rumen-protected methionine supplementation on the performance of high production dairy cows in the tropics

PLOS ONE

Dear Dr. Collao-Saenz,

Thank you for submitting your manuscript to PLOS ONE. After careful consideration, we feel that it has merit but does not fully meet PLOS ONE’s publication criteria as it currently stands. Therefore, we invite you to submit a revised version of the manuscript that addresses the points raised during the review process.

We look forward to receiving your revised manuscript.

Kind regards,

Juan J Loor

Academic Editor

PLOS ONE

We note that one or more of the authors are employed by a commercial company: Schwab Consulting LLC, Boscobel, WI, USA

(2) Please also provide an updated Competing Interests Statement declaring this commercial affiliation along with any other relevant declarations relating to employment, consultancy, patents, products in development, or marketed products, etc.  

Reviewers' comments:

Reviewer's Responses to Questions

**Comments to the Author**

1. Is the manuscript technically sound, and do the data support the conclusions?

Reviewer #1: Yes

Reviewer #2: Yes

2. Has the statistical analysis been performed appropriately and rigorously? 

Reviewer #1: Yes

Reviewer #2: Yes

3. Have the authors made all data underlying the findings in their manuscript fully available?

Reviewer #1: Yes

Reviewer #2: Yes

4. Is the manuscript presented in an intelligible fashion and written in standard English?

Reviewer #1: Yes

Reviewer #2: Yes

5. Review Comments to the Author

Reviewer #1: General comments:

The manuscript aims to evaluate the supplementation of smartamine (RP methionine) on milk performance and blood AA profile in mid-lactating dairy cows of a commercial dairy farm located in a tropic climate.

The manuscript adds some relevant information regarding the raising importance topic in the formulation of diet with RP AA supplementation in dairy cows, particularly considering the climate and the fact that the experiment was conducted in a commercial farm.

However, there are several concerns raised up from the reviewing process. First of all, the clearness of the methods adopted and data about cows at the beginning of the trial.

The discussion section is sometime confusing, and authors speculated heavily on some aspect without the support of their own data.

I suggest also to switch the manuscript as “short communication”.

Specific comments

L15: What do you mean with adequate? It is not a correct form to state it.

L19: Please put the name of the methionine group in the brackets (I assume SM).

L21: If the experiment lasts 10 wk, why did authors collect blood samples at 30 d? BCS and BW was measured at the end though.

L23-27: For some variables, authors state the exact p value and for others they indicate with ≤. Please be consistent. In addition, I assume that authors have only a p value (CON vs SM, overall) for ECM, milk protein yield, milk protein %, and casein, so why do not they use the exact p value?

L73-81: Not only for the mechanisms pointed out, but also for the antioxidant system (taurine and glutathione). Please re-phrase this part taking into account these functions linked to the methionine cycle and transsulfuration pathway.

L81: Please, add the reference.

L87: In the authors’ hypothesis should be also stated the potential changes in the plasma AA profile.

L91: It is not correct “early lactation cows”. Authors mentioned the first half and since they started on average at 65 DIM and concluded 10 wk after, it cannot be considered early lactation. Re-phrase according to this comment.

L92: “correct amount” to reach the ratio required for the experiment objective. There is not an absolute right amount of Met supplementation.

L102-105: Since authors stressed out the tropical environment in the hypothesis/objective, it would be good to add some info on the climatic condition during the trial. It seems important to have these data, even though there is not a comparison with other climes such as North America and Europe.

L102: equipped is not the right word.

L104: Please make clearer this point.

L107-114: Please put the average of milk yield, bw, bcs and DIM at the beginning of the experiment in the table with the two groups separated.

L118-122: Authors have already these data in table, thus it does not need to be reported again.

L123-124: It is not clear how authors obtained the 14 g/d of true Met starting from 23 g/d of Smartamine. Did the authors conducted in vitro/in vivo experiment to establish the true by-pass Met? If not, it is better to report the Samrtamine addition as % of DM and then specifying the % of Met provided by Smartamine, as authors stated in the L125.

Table 1: I guess there is a mistake indicating “0.09% of DMI”. Probably authors were supposed to write “0.09% of DM”. If authors specify DMI lead to a mistake because it assumes that the 0.09% refers to the true DMI. Or referring as 0.09% of predicted DMI. Up to authors.

Was the Met provided as top-dress or mixed in the TMR?

Table 2: The authors could even eliminate the table since it does not give relevant support at the paper. Those values come up from prediction equations, and authors do not have DMI data.

L163: Milk samples were collected from one milking or composited from the 3 milking of the same day?

L175: 12 cows for blood samples up to 38 cows totally involved in each group seems a little number of subjects. Could authors explain why?

Why only at 30 days after starting treatment and not at the end or both?

L177: Why 6 hours after delivering the first TMR?

L236: Individual DMI? I guess not, so I suggest stating clearly that is an average of the group-fed cows in each treatment. According to this, I suggest also not to speculate too much.

L281: Very interesting result but I would like to see data during the 2-wk of adaptation according to the treatment (SM and CON) and if the covariate inclusion was significant in the model.

L308-313: How can it be attached to the previous paragraph discussing the milk lactose? It is in contrast, firstly, and I do not see the link between these two aspects considering also the lack of data on blood glucose concentration.

However, it is not correct talking about inflammatory state because authors do not have data on blood biomarkers. I would eliminate this part.

L349-353: Unfortunately, authors do not have data to discuss about this. Consider eliminating and I suggest the authors to keep the discussion according to their data.

Reviewer #2: This is a well written manuscript with concise hypothesis and objectives. The results seem to be a straightforward interpretation and, for the most part, are in line with prior data on this topic. The main concern of this reviewer is the unclear collection, data analysis, and interpretation of individual or collective (e.g., pen) DMI. This needs to be clarified before this manuscript can be suitable for publication.

Lines 104-105 “A total mixed ration (TMR) was offered…”

Line105 Explain how this was calculated. Was this done by total TMR offered to the pen? Were the 38 cows per treatment kept on single separate pens, or were there multiple pens?

Lines 109-110 What about previous lactation yield or 305ME corrected lactation yield?

Line 117 Please correct this “(indicate what you used)”.

Lines 119-122 Please reference Table 1 here.

Line 146 “Predicted DM intake” instead of “average DM intake”. Also, what about milk yield, milk composition, BW, and BCS? All these parameters are required for NRC to make predictions on requirements for RDP, RUP, and MP.

Line 198 Delete “and cow”

Line 219 “RPMet”

Lines 236-237 Where are these data coming from? From the Materials and Methods section, it is not clear how the authors collected individual DMI.

Line 270 Provide references.

Line 273 Which AA? The authors should better this theory.

Lines 277-278 How does this relate to a more efficient use of other absorbed AA?

Lines 305-306 Please clarify, do the authors meant that the mammary gland has the flexibility to use other substrates for gluconeogenesis (e.g., BHB and lactate)? If so, provide references.

Line 307 Not clear what is “it”?

Line 318 Not clear what the authors meant by “fractional protein synthesis”.

Lines 318-319 Provide reference.

Lines 319-321 Provide reference.

Line 324 “maximum protein yield”

Lines 353-355 Provide reference.

Line 368 “(P< 0.05)”

Line 401 This reviewer suggests changing this to “Met was a limiting nutrient…” Since this experiment was not designed to compare Met against other nutrients, this statement needs to be reworded to be in agreement with the experimental design and results.

Line 406 This reviewer suggest to double-check these uM units here and in Table 4. This unit seem to be “uM” instead of “uM/mL”.

6. PLOS authors have the option to publish the peer review history of their article (what does this mean?). If published, this will include your full peer review and any attached files.

Reviewer #1: No

Reviewer #2: No

---

## [Author Response · Author response to Decision Letter 0]

2 Mar 2021

Most of the Reviewer's suggestions and comments were accepted and revised in the text and are detailed in the “Response to Reviewers” file

---

## [Decision Letter · Decision Letter 1]

19 Apr 2021

Effects of rumen-protected methionine supplementation on the performance of high production dairy cows in the tropics

PONE-D-20-37159R1

Dear Dr. Collao-Saenz,

We’re pleased to inform you that your manuscript has been judged scientifically suitable for publication and will be formally accepted for publication once it meets all outstanding technical requirements.

Kind regards,

Juan J Loor

Academic Editor

PLOS ONE

Additional Editor Comments (optional):

Reviewers' comments:

Reviewer's Responses to Questions

**Comments to the Author**

1. If the authors have adequately addressed your comments raised in a previous round of review and you feel that this manuscript is now acceptable for publication, you may indicate that here to bypass the “Comments to the Author” section, enter your conflict of interest statement in the “Confidential to Editor” section, and submit your "Accept" recommendation.

Reviewer #1: All comments have been addressed

2. Is the manuscript technically sound, and do the data support the conclusions?

Reviewer #1: Yes

3. Has the statistical analysis been performed appropriately and rigorously? 

Reviewer #1: Yes

4. Have the authors made all data underlying the findings in their manuscript fully available?

Reviewer #1: Yes

5. Is the manuscript presented in an intelligible fashion and written in standard English?

Reviewer #1: Yes

6. Review Comments to the Author

Reviewer #1: (No Response)

7. PLOS authors have the option to publish the peer review history of their article (what does this mean?). If published, this will include your full peer review and any attached files.

Reviewer #1: **Yes: **Vincenzo Lopreiato

---

## [Editor Report · Acceptance letter]

21 Apr 2021

PONE-D-20-37159R1 

Effects of rumen-protected methionine supplementation on the performance of high production dairy cows in the tropics 

Dear Dr. Collao-Saenz:

I'm pleased to inform you that your manuscript has been deemed suitable for publication in PLOS ONE. Congratulations! Your manuscript is now with our production department. 

Kind regards, 

on behalf of

Dr. Juan J Loor 

Academic Editor

PLOS ONE